# Surgical Implications of Advanced Low-Grade Serous Ovarian Cancer: Analysis of the Database of the Tumeurs Malignes Rares Gynécologiques Network

**DOI:** 10.3390/cancers14092345

**Published:** 2022-05-09

**Authors:** Hélène Bonsang-Kitzis, Nabilah Panchbhaya, Anne-Sophie Bats, Eric Pujade-Lauraine, Patricia Pautier, Charlotte Ngô, Marie-Aude Le Frère-Belda, Elsa Kalbacher, Anne Floquet, Dominique Berton-Rigaud, Claudia Lefeuvre-Plesse, Michel Fabbro, Isabelle Ray-Coquard, Fabrice Lécuru

**Affiliations:** 1Gynecological and Breast Surgery and Cancerology Center, RAMSAY-Générale de Santé, Hôpital Privé des Peupliers, 75013 Paris, France; h.kitzisbonsang@yahoo.fr (H.B.-K.); charlotte.ngo2@gmail.com (C.N.); 2GINECO/TMRG Network, 75008 Paris, France; epujade@arcagy.org (E.P.-L.); patricia.pautier@gustaveroussy.fr (P.P.); marie-aude.le-frere-belda@aphp.fr (M.-A.L.F.-B.); ekalbacher@gmail.com (E.K.); a.floquet@bordeaux.unicancer.fr (A.F.); dominique.berton-rigaud@ico.unicancer.fr (D.B.-R.); c.lefeuvre-plesse@rennes.unicancer.fr (C.L.-P.); michel.fabbro@icm.unicancer.fr (M.F.); isabelle.ray-coquard@lyon.unicancer.fr (I.R.-C.); 3Department of Obsterics and Gynecology, Hôpital Lariboisière, Assistance Publique—Hôpitaux de Paris, 75020 Paris, France; nabilah.panchbhaya@aphp.fr; 4Department of Breast and Gynecological Surgical Oncology, Hôpital Européen Georges Pompidou, Assistance Publique—Hôpitaux de Paris, 75015 Paris, France; anne.sophie.bats@aphp.fr; 5School of Medicine, Université de Paris, 75005 Paris, France; 6Women Cancer Center and Clinical Research, Hôpital Hôtel-Dieu, Assistance Publique—Hôpitaux de Paris, 75006 Paris, France; 7Department of Medical Oncology, Institut Gustave Roussy, 94000 Villejuif, France; 8Department of Pathology, Hôpital Européen Georges Pompidou, Assistance Publique—Hôpitaux de Paris, 75015 Paris, France; 9Department of Medical Oncology, Hôpital Jean Minjoz, 25000 Besançon, France; 10Department of Medical Oncology, Institut Bergonié, 33000 Bordeaux, France; 11Department of Medical Oncology, Institut de Cancérologie de l’Ouest-René Gauducheau, 44000 Saint Herblain, France; 12Department of Medical Oncology, Centre Eugène Marquis, 35000 Rennes, France; 13Department of Medical Oncology, Institut Régional du Cancer de Montpellier, 33000 Montpellier, France; 14Department of Medical Oncology, Centre Léon Bérard, 69000 Lyon, France; 15School of Medicine, Claude Bernard University, 69000 Lyon, France; 16Department of Breast, Gynecology and Reconstructive Surgery, Institut Curie, 75005 Paris, France

**Keywords:** low-grade serous ovarian cancer, advanced stage, surgery, neoadjuvant chemotherapy, survival

## Abstract

**Simple Summary:**

Low-grade serous carcinoma is a recent entity. The surgical management of advanced stages is modeled on that of high-grade tumors, with the use of neoadjuvant chemotherapy in the case of carcinosis not amenable to complete primary resection. We retrospectively analyzed data from the French national network dedicated to rare gynecologic tumors. We compared disease extension, surgical characteristics, postoperative course and survival after primary surgery vs. interval debulking. Carcinosis was more extended in the case of neoadjuvant chemotherapy. However, chemotherapy did not reduce surgical complexity, nor late postoperative morbidity. Surprisingly, progression-free and overall survival were similar after complete macroscopic or minimal resection (residuals < 2.5 mm). Survival was similar in the case of residuals ≥2.5 mm or more and nonoperated patients. Neoadjuvant chemotherapy does not improve the resectability of advanced low-grade serous cancers. Primary cytoreduction with complete or with minimal residuals should be preferred when feasible.

**Abstract:**

The surgical specificities of advanced low-grade serous ovarian carcinoma (LGSOC) have been little investigated. Our objective was to describe surgical procedures/complications in primary (PDS) compared to interval debulking surgery (neoadjuvant chemotherapy and interval debulking surgery, NACT-IDS) and to assess the survival (progression-free (PFS) and overall survival (OS)) in patients with advanced LGSOC. We retrospectively analyzed advanced LGSOC from a nationwide registry (January 2000 to July 2017). A total of 127 patients were included (48% PDS and 35% NACT-IDS). Peritoneal carcinomatosis was more severe (*p* = 0.01 to 0.0001, according to sites), surgery more complex (*p* = 0.03) and late postoperative morbidity more frequent (*p* = 0.03) and more severe in the NACT-IDS group. PFS and OS were similar in patients with CC0 and CC1 residual disease after PDS or IDS. Prognosis was poorest for NACT-IDS patients with CC2/CC3 resection (PFS: HR = 2.31, IC95% (1.3–4.58); *p* = 0.005; OS: HR = 4.98, IC95% (1.59–15.61); *p* = 0.006). NACT has no benefit in terms of surgical outputs in patients with advanced LGSOC. Patients with complete resection or minimal residual disease (CC0 and CC1) have similar prognoses. On the other hand, patients with CC2 and more residual disease have similar survival rates compared to nonoperated patients. Primary cytoreduction with complete or with minimal residuals should be preferred when feasible.

## 1. Introduction

Low-grade serous ovarian carcinoma (LGSOC) is a recent entity, identified by Malpica et al. [1] and by Kurman and Shih [2]. LGSOC accounts for less than 10% of serous ovarian cancers and 5 to 8% of ovarian cancers [3], which classifies this subtype as a rare disease.

Several differences have been noticed between low- and high-grade cases. At the molecular level, mitogen-activated protein kinase (MAPK) and the PI3K/AKT/mTOR pathways play a prominent role in the pathogenesis of both serous tumors of low malignant potential and low-grade serous carcinomas, which can be a precursor [4,5]. Clinically, patients are younger, have slightly longer progression-free survival (PFS) and have longer overall survival (OS) than high-grade cancers [3,6]. These features are observed despite a predominance of advanced stages at diagnosis and frequent residual disease at completion of primary treatment.

LGSOC has been treated in the same manner as high-grade cancers for a long time, but the surgical specificities of advanced LGSOC have been little investigated. For high-grade serous cancers, complete removal of gross disease at primary debulking surgery (PDS) is unanimously recognized as the main prognostic factor and thus the main goal of primary surgery [7,8]. This has led to the development of radical or ultraradical surgical procedures, aiming at observing no residual macroscopic disease in more than 70–80% of operated patients in trained centers. Conversely, neoadjuvant chemotherapy + interval debulking surgery (NACT + IDS) is proposed for patients not amenable to primary complete cytoreduction or who could not tolerate an aggressive surgery [9]. The applicability of this strategy to LGSOC is questionable due to the poor chemosensitivity of this tumor (response rate of 4 to 25%) [10,11].

Our objective was to compare the surgical characteristics of primary compared to interval debulking surgery in patients with advanced LGSOC. The secondary objective was to assess the survival outcome after primary versus interval surgery. The main endpoint was surgical complexity/morbidity. The secondary endpoints were PFS and OS.

## 2. Material and Methods

### 2.1. Population

We performed a retrospective study on the prospectively maintained database of the “Tumeurs Malignes Rares Gynécologiques” network [12]. This network is labeled by the French National Cancer Institute (INCA) and operated by the ARCAGY-GINECO cooperative group. It aims to offer pathological review by experts, as well as treatment advice, with three national multidisciplinary meetings (Lyon, Paris, Villejuif) to edit guidelines and promote epidemiologic and clinical studies on rare malignant gynecologic tumors.

Inclusion criteria were stage FIGO III or IV LGSOC, confirmed by expert pathological review and scheduled for surgery (primary debulking surgery or interval debulking surgery after neoadjuvant chemotherapy). Exclusion criteria were patients without pathological review or another final diagnosis. The study period ranged from January 2000 to July 2017 (cutoff date for analysis).

Clinical characteristics, imaging results, operative reports, pathology reports, hospital reports and follow-up were computed (Excel 15.33). Carcinosis extension was scored using the Peritoneal Cancer Index [13]. Surgical complexity was described using the Pomel and Dauplat classification [14] and the Aletti classification [15]. Pre- and postoperative complications were graded according to the Common Terminology Criteria for Adverse Events (CTCAE 4.0) [16]. Complications were defined as early (before 30 days) or late (after 30 days). Residual disease was rated according to the Completeness of Cancer Resection (CCR) score [17]. This score can be characterized by the amount of the largest remaining lesion after debulking surgery (patients with no visible disease (CC0), disease in which the remaining nodule is <2.5 mm (CC1), disease in which the remaining nodule is ≥2.5 mm to ≤2.5 cm (CC2) and disease in which the remaining nodule is >2.5 cm (CC3)).

Therapeutic procedures and their sequences were based on the guidelines of the National Network and were discussed in multidisciplinary meetings (surgery, chemotherapy, hormonal therapy or targeted therapy). Patients were judged not eligible for primary debulking surgery if complete resection was deemed not probable at the laparoscopic evaluation. Chemotherapy combined paclitaxel (175 mg/m^2^) and platinum (AUC 5) in 98.1% of patients. Follow-up was based on clinical examination, serum CA125 levels and CT scan every 4 months for 2 years, every 6 months for 3 years, and then annually after the fifth year.

Recurrence was defined by an elevation of CA125 and/or radiological abnormality (computerized tomography (CT) scan of thorax, abdomen and pelvis and/or positron emission tomography (PET) CT).

### 2.2. Statistical Analysis

Descriptive analysis was presented in terms of frequencies for qualitative variables or medians and associated range for quantitative variables. The cutoff date for analysis was July 2017. The whole population was described and then divided into a primary debulking surgery group (PDS) and neoadjuvant chemotherapy, followed by an interval debulking surgery group (NACT-IDS). PDS and NACT-IDS populations were compared using the chi-square test, the chi-square test with Yates’ correction, Fisher’s exact test or the ANOVA test when appropriate.

Progression-free survival (PFS) was defined by the time interval between pathological diagnosis and recurrence, progression or death. Overall survival (OS) was defined by the time interval between pathological diagnosis and death. Survival was described using Kaplan–Meier estimation, and a comparison between survival curves was performed with the log-rank test. Estimation of hazard ratios (HRs) and their associated 95% confidence interval (CI) was carried out using the Cox proportional hazard model. Multivariate analysis was not performed due to the small number of events. Surgical outcomes in terms of PDS and IDS were primary endpoints for PFS and OS.

A threshold of 0.05 was used for significance levels. Statistical analyses were performed with R version 3.5.0 (http://cran.rproject.org, accessed on 18 March 2022) using the following packages: glm, survival and ggplot2.

### 2.3. Ethical Approval

The website with the database created to improve the management of the cohort of women with rare gynecological malignant tumors was endorsed by the French authorities (“Comité Consultatif sur le Traitement de l’Information en matière de Recherche dans le domaine de la Santé” (CCTIRS), Authorization Numbers: 09.342 and 09.342bis; “Comité National Informatique et Liberté” (CNIL), Authorization Number: 909454) for registration of adult patients. Further notification was formally made to the CCTIRS for patients over 15 years old in 2014. The complete organization, including the database and multidisciplinary tumor boards, was also endorsed and labeled by the National Cancer Institute in 2011, 2014 and 2019. The follow-up of patients and the creation of a database dedicated to rare cancers are missions issued by the French authorities through Plan Cancer 2009–2013. According to French law, all patients are informed by their physicians about the network’s goals as a centralized platform dedicated to rare gynecological cancer management.

The multidisciplinary tumor board for cancer management (of initial treatment and at relapse) is required by law in France (Plan Cancer 2003–2007). Thus, the recommendation to discuss rare ovarian tumors via our dedicated tumor boards is directly in accordance with French legacy. This study was specifically approved by the Scientific Committee of the “Tumeurs Malignes Rares Gynécologiques” network. Patients included in the network signed informed consent. The institutions that have registered patients and their medical records in this national database are listed below.

For specific biological research purposes, patients can sign the institutional informed consent, as well as the specific informed consent for rare ovarian tumors, and both are considered equally effective for further studies on biological samples.

All data were fully anonymized. Patients’ medical records were accessed between April 2016 and July 2017.

## 3. Results

One hundred and twenty-seven patients with stage III/IV LGSOC from 25 centers were finally included (Figure 1). Of note, 70/281 (25%) patients with an initial diagnosis of LGSOC were excluded after histological review. The main characteristics of our population are given in Table 1. Most patients had stage IIIc/IV disease (120/127, 94.5%). Sixty-one patients had a PDS (48%) and 44 (34.6%) NACT-IDS. Eighteen patients (14.2%) did not have debulking surgery at all. These patients were significantly older (median age of 70) and often had stage IV disease.

### 3.1. Primary Objective: Comparison of Surgical Characteristics between Primary and Interval Debulking Surgery

The two groups were not significantly different for age at diagnosis and the initial FIGO stage. A comparison of the initial disease extension, interventions and surgical output is given in Table 2. Carcinomatosis extension was more severe in cases with interval surgery compared to primary surgery. The Peritoneal Cancer Index was significantly higher (median of 14 (2–33) for NACT-IDS versus 6 (3–24) for the PDS group, *p* = 0.03). These patients had significantly more frequent digestive, diaphragmatic and upper-abdominal peritoneum involvement (*p* = 0.01; *p* = 0.003; *p* = 0.0001, respectively). Surgery was more complex after NACT than at PDS. The distribution of the Pomel and Dauplat classification, as well as the Aletti score, was significantly different between groups. In particular, bowel resection and diaphragmatic peritoneum stripping were significantly more frequent after NACT (54.8% and 58.1% versus 34.4% and 36.7%, respectively). At the end, surgery completeness was similar in both groups. The rate of patients without macroscopic residuals or with minimal residuals (CC0/CC1) was 90% after primary debulking and 86% after interval debulking (*p* = 0.59). Pelvic and/or para-aortic lymphadenectomy was performed, respectively, in 78.3% and 76.2% of patients.

Intraoperative morbidity was frequent and consisted mainly of CTCAE grade 1 and 2 events, without differences between groups (*p* = 0.1). Early postoperative complications were similar between groups (*p* = 0.7) (Table 3). The most frequent intraoperative and early postoperative complications found were diaphragmatic injuries (14.7%), digestive injuries (7.8%) and hemorrhagic complications (6.2%). Late postoperative complications were significantly more frequent (*p* = 0.03) in the NACT-IDS group. However, the rate of grade 3 and 4 CTCAE complications was similar between the two groups (57.1% and 58.8%, respectively). The most frequent late postoperative complications concerned wounds, including hematomas, scar disunions and herniations of the abdominal wall (12.5%). Digestive complications (10.7%, 7% of which digestive occlusion) and lymphoceles (5.4%) were also recorded. Significantly more frequent blood transfusion was reported after NACT compared to PDS (*p* = 0.018). In summary, surgical complexity and intra-/postoperative complications were not reduced after neoadjuvant chemotherapy.

### 3.2. Secondary Objective: Comparison of Survival

Concerning postoperative treatment, a median of 6 (range from 1 to 12) cycles of chemotherapy were given in the PDS group and 3 (range from 1 to 8) in the NACT group (*p* = 9.6 ^10–9^) (a median of 4 cycles of chemotherapy were given before surgery (1–4) in patients receiving NACT). Bevacizumab was significantly more frequently given in the NACT-IDS group (34.1%, *p* = 0.01) and for patients with CC0/CC1 score after debulking surgery (CC0: 25%, CC1: 25%, CC2: 20%, CC3 0%, *p* = 0.003). Eight patients (7.6%) received adjuvant hormonal therapy (six in the PDS and two in the NACT-IDS group) (Table 4, Table 5).

Median follow-up was 27 months (range: 11–67 months). A total of 72 patients (56.7%) had a recurrence or progression. Among these patients, 47.2% (34/72) had disease progression (35% in PDS and 65% in NACT-IDS groups), and 52.8% (38/72) had recurrence (57.1% in PDS and 42.9% in NACT-IDS groups).

Median PFS was 45 months (7–140) after primary surgery vs. 28 months (2–120) after NACT-IDS (*p* = 0.04). Survival according to timing of surgery and residual disease is given in Figure 2. We observed three main results:(1)Patients with complete macroscopic resection and patients with minimal residual disease (CC1) had similar survival rates (HR = 0.81, IC95% (0.33–1.97)).(2)PFS was similar after PDS or NACT-IDS in patients with CC0/CC1 resection (HR = 1.64, IC95% (0.88–3.04); *p* = 0.12).(3)Patients with macroscopic residual disease (CC2 and more) had the worst prognosis (HR = 2.31, IC95% (1.3–4.58); *p* = 0.005). These patients had a similar outcome to that of nonoperated patients (compared to CC0 patients as reference: HR = 3.68, IC95% (1.44–9.39); *p* = 0.006 and HR = 3.96, IC95% (1.93–8.14); *p* = 0.0002, for CC2 and more or nonoperated patients, respectively) (Figure 2).

Estrogen receptor expression was an indicator of good prognosis in the overall population (HR = 0.41, IC95% (0.17–0.89), *p* = 0.02) and particularly for patients with PDS (HR = 0.04, IC95% (0.001–0.63), *p* = 0.001). A positive peritoneal cytology (*p* = 0.67), metastatic pelvic or para-aortic nodes (*p* = 0.06), perioperative complications (*p* = 0.99), early postoperative complications (*p* = 0.36), late postoperative complications (*p* = 0.71), progesterone receptor expression (*p* = 0.43), adjuvant chemotherapy (*p* = 0.45), use of bevacizumab (*p* = 0.93) and use of hormonal therapy (*p* = 0.12) had no impact on PFS (Table 3).

Median OS was 130 months (7–163) after primary surgery vs. 82 months (7–120) after NACT-IDS (*p* = 0.003).

OS was significantly better in patients with primary cytoreduction (HR = 0.4, IC95% (0.18–0.9); *p* = 0.027). We observed four main results:(1)Patients who achieved CC0/CC1 resection had a similar OS after PDS or IDS (HR = 1.92, IC95% (0.78–4.71); *p* = 0.15).(2)In the PDS group, the residual disease after surgery had no significant impact on OS (CC2/CC3 versus CC0/CC1 as reference: HR = 1.19, IC95% (0.23–6.08); *p* = 0.8).(3)After NACT-IDS, CC2/CC3 patients had a significantly worse prognosis compared to CC0/CC1 patients (HR = 4.98, IC95% (1.59–15.61); *p* = 0.006).(4)Survival of NACT-IDS CC2 or more patients was not significantly different from those without any surgery (compared to CC0 patients as reference: HR = 7.65, IC95% (2.42–24.15); *p* = 0.0005 and HR = 5.51, IC95% (2.02–15.01); *p* = 0.0008, for CC2 and more or not operated patients respectively) (Figure 2).

Estrogen receptor expression was an indicator of better OS for overall population (HR = 0.23, IC95% (0.08–0.64), *p* = 0.005) and particularly for patients with PDS (HR = 0.05, IC95% (0.001–0.79), *p* = 0.003). Age > 45 (HR = 2.32, IC95% (1.03–5.32); *p* = 0.04), use of neoadjuvant chemotherapy (HR = 2.64, IC95% (1.37–5.06); *p* = 0.003) and use of a hormonal therapy (HR = 2.81, IC95% (1.15–6.88); *p* = 0.024) were associated with a significantly shorter OS (Table 6).

## 4. Discussion

LGSOC has been individualized from high-grade serous cancers for 20 years. It has been treated in the same manner as HGSOC for a long time. However, poor response to chemotherapy has been shown [10,18,19], and hormonal therapy could be a valuable option as first-line adjuvant therapy [20]. The specificity of surgery for LGSOC has received little attention until now. For HGSOC, the goal of complete resection at PDS and the option of NACT + IDS for patients judged unresectable or inoperable are shared among the community of gynecologic oncologists. However, for LGSOC, no specific data exist on this issue, and few studies have been performed, as the number of cases is low. However, this question is important, since a majority of patients are diagnosed at an advanced stage, and frequent residual disease has been reported [3,21].

We decided to investigate the outcome of surgery for advanced LGSOC, in terms of surgical morbidity and survival, with a comparison between primary and interval surgery after NACT. The originality of this study was to use the data of a national network on rare gynecologic cancers, providing more cases and a more realistic view than a single referent center (real-life data). We recorded two major results: (1) NACT had no favorable effect on surgery in these patients, since complexity of surgery (*p* = 0.03), rate/severity (*p* = 0.03 and *p* = 0.05, respectively) of late postoperative complications were significantly higher than after PDS without benefit in terms of residual disease and survival; (2) the cutoff for surgical resection is different in LGSOC, since patients with CC0 and CC1 score had similar survival (PFS: HR = 0.81, IC95% (0.33–1.97) and OS: HR = 1.12, IC95% (0.33–3.83) for CC1 resections compared to CC0). This finding was observed after PDS, as well as after NACT-IDS.

Our population was similar to that previously described in the literature in terms of median age [3,19], stage at diagnosis [21] and rate of complete resection at PDS. Our main difference is the rate of NACT-IDS (34.6%) reflecting French practice for surgery of ovarian cancer during the study period.

We observed a high rate of complex surgery at the time of primary surgery, and even more at the time of interval debulking. This result was obtained using the Pomel and Dauplat classification, as well as the Aletti score. It is interesting to note that radical and ultraradical surgery was more frequent in patients with LGSOC than in patients with high-grade cases in our center, as described in the literature [22,23]. This was explained by a more severe initial presentation with significantly more frequent diaphragm, liver, upper-abdominal peritoneum and bowel involvements. This also confirms the poor response of this disease to preoperative chemotherapy, contrary to HGSOC. Morbidity was high in this advanced LGSOC population, but was similar between groups, except for late complications, which were significantly more frequent after interval debulking. This emphasizes the fact that indications of NACT should not be the same for LGSOC and high-grade cancers, since the benefit of NACT reported for high-grade cancers is not observed in LGSOC cases [24]. A recent paper (25) also addressed the surgical characteristics of LGSC and HGSC. Jonhson et al. also reported similar results. In this series, 65% of patients with LGSC were proposed to PDS vs. 20% in HGSC. Complete reduction (CC0 and CC1) was obtained in 89.2% of patients. A moderate or high surgical complexity score was reported in 59.5% and 8% of cases. They also reported longer operations, more blood transfusions, longer hospital stays and more postoperative complications in LGSC when compared to HGSC. Similarly, to our results, the best survival was obtained after PDS and complete resection (CC0/CC1). One unreported finding was the benefit of aggressive surgery (SCS ≥ 4) on survival. This suggests that for LGSC, the classical blocking points used in the case of HGSC should not be used, and a specific surgical approach is necessary. This series also reports no survival advantage of (neo)adjuvant chemotherapy (25).

Survival rates in this series were in accordance with published data [3,11,18,25]. We recorded similar PFS and OS rates for patients with complete resection or minimal residual disease (CC0 and CC1) in the PDS and NACT-IDS groups, as well as similar OS for all classes of residual disease in the PDS population. Gershenson et al. previously reported similar results concerning OS in two series of 112 and 350 patients [18,21]. Fader et al. [6] published contradicting results, with better PFS and OS in patients having microscopic residual disease. Grabowski et al. published data on 145 LGSOC patients within AGO studies. They came to the conclusion that R0 and R < 1 cm give similar PFS and OS rates and worse PFS and OS rates for resections R > 1 cm [11]. However, direct comparison of our results to previous results is questionable, since the cutoff for residual disease was not the same (inferior or superior to 2cm for Gershenson et al. [21], 0.1–1 cm, superior to 1 cm for Fader et al. [6] and >1 cm for Grabowski et al. [11]). However, our results and those from Gershenson et al. and Grabowski et al. could be a strong argument to support primary surgery, even if complete macroscopic resection is not certain. This study also suggests that for low-grade tumors, the CCR score in four classes (CC0/CC1/CC2/CC3) could be revised by merging CC0/CC1 groups. Finally, surgery is questionable after NACT in the case of residual disease ≥ CC2, since morbidity is increased, and survival rates are similar to those of nonoperated patients.

We present here the unique study coming from a nationwide registry. Multicentricity, a long study period and a high number of patients enable obtaining robust results. Such large series are uncommon for rare diseases. Pathological review is also an important quality criterion to ensure that all cases are indeed LGSOC. However, some limitations are obvious. The retrospective character of our analysis despite a prospective inclusion of LGSOC patients, the potential selection bias of patients with worse disease triaged to NACT, limited follow-up for LGSOC with a frequently indolent clinical course and missing data could limit some analyses (the Peritoneal Cancer Index was not systematically recorded, and minor complications could be underestimated).

## 5. Conclusions

Treatment of advanced-stage LGSOC is a difficult and challenging situation for clinicians. The surgical approach of LGSOC should be different from that of HGSOC.

Neoadjuvant chemotherapy does not provide a surgical advantage in this disease. Patients with complete resection or minimal residuals have a similar survival.

Primary debulking surgery (with maximal effort) should be favored, even in case of minimal residual.

## Figures and Tables

**Figure 1 cancers-14-02345-f001:**
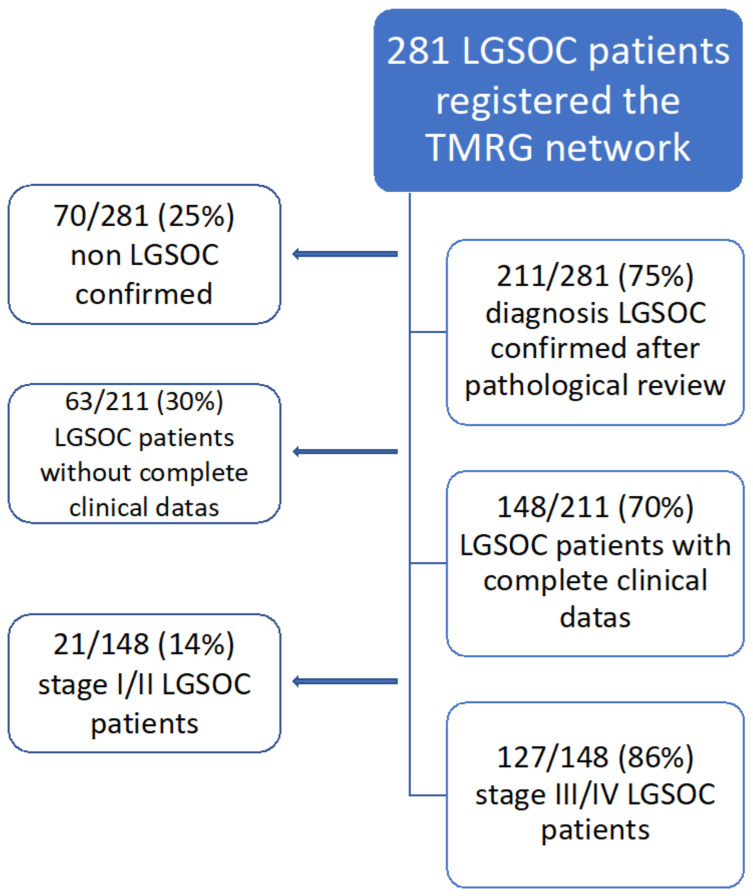
Flowchart of the study.

**Figure 2 cancers-14-02345-f002:**
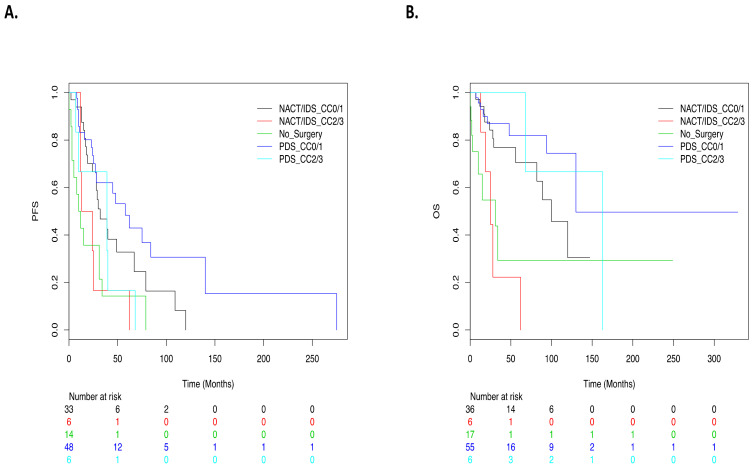
Survival curves. (**A**) Progression-free survival according to time of debulking surgery and Completeness of Cancer Resection (CCR) score. (**B**): Overall survival according to time of debulking surgery and Completeness of Cancer Resection (CCR) score. Five groups: no surgery in green, primary debulking surgery with optimal or incomplete resection (PDS-CC0 or CC1 in dark blue, PDS-CC2 or CC3 in light blue), interval debulking surgery with optimal or incomplete resection (NACT/IDS-CC0 or CC1 in dark blue, NACT/IDS-CC2 or CC3 in red).

**Table 1 cancers-14-02345-t001:** Comparison of the PDS and NACT-IDS populations (disease extension).

	PDS *n* (%) or Median (IQ Range)	NACT-IDS *n* (%) or Median (IQ Range)	Total *n* (%) or Median (IQ Range)	*p* (Chi2, Yates, Fisher or Student)
**Age**	54 (37–62)	55 (42–69)	54 (38–68)	0.4
**Body mass index**	23 (19–27)	24 (22–28.5)	24 (21–28)	0.4
**Postmenopausal**	35 (59.3)	22 (55.0)	71 (55.9)	0.98
**Initial CA125 (UI.L-1)**	122.5 (29.75–433.5)	355.5 (156.2–997.2)	273.5 (103.8–594.8)	0.05
**FIGO stage**				
IIIA	2 (3.1)	0	2 (1.6)	
IIIB-IV	63 (96.9)	61 (100)	124 (98.4)	0.5
**PCI**	6 (3–24)	14 (2–33)	8 (3–33)	0.03
**Digestive involvement**				
no	39 (66.1)	17 (41.5)	56 (56.0)	
yes	20 (33.9)	24 (58.5)	44 (44.0)	0.0146
**Diaphragmatic involvement**				
no	41 (69.5)	13 (32.5)	54 (54.5)	
yes	18 (30.5)	27 (67.5)	45 (45.5)	0.003
**Liver capsule involvement**				
no	57 (96.6)	32 (82.1)	89 (90.8)	
yes	2 (3.4)	7 (17.9)	9 (9.2)	0.037
**Splenic involvement**				
no	71 (91.0)	32 (84.2)	103 (88.8)	
yes	7 (9.0)	6 (15.8)	13 (11.2)	0.5796
**Upper abdomen peritoneum involvement**				
no	39 (65.0)	10 (25.0)	49 (49.0)	
yes	21 (35.0)	30 (75.0)	51 (51.0)	0.0001

PDS: primary debulking surgery NACT-IDS: neoadjuvant chemotherapy and interval debulking surgery.

**Table 2 cancers-14-02345-t002:** Comparison of the PDS and NACT-IDS populations (surgical characteristics).

	PDS *n* (%)	NACT-IDS *n* (%)	Total *n* (%)	*p* (Chi2, Yates or Fisher)
**Pomel and Dauplat Classification**				
Standard surgery	24 (35.8)	12 (27.2)	36 (32.4)	
Radical surgery	18 (26.9)	5 (11.4)	23 (20.7)	0.03
Ultra-radical surgery	25 (37.3)	27 (61.4)	52 (46.8)	
**Aletti score**				
Low complexity	8 (12.9)	4 (8.9)	12 (10.8)	
Intermediate complexity	37 (59.7)	16 (35.6)	57 (51.4)	0.001
High complexity	17 (27.4)	25 (55.6)	17 (37.8)	
**Digestive resection**				
no	40 (65.6)	19 (45.2)	59 (57.3)	
yes	21 (34.4)	23 (54.8)	44 (42.7)	0.04
**Posterior pelvectomy**				
no	42 (68.9)	21 (51.2)	63 (61.8)	
yes	19 (31.1)	20 (48.8)	39 (38.2)	0.07
**Diaphragmatic stripping**				
no	38 (63.3)	18 (41.9)	56 (54.4)	
yes	22 (36.7)	25 (58.1)	47 (45.6)	0.03
**Pelvic or para-aortic lymphadenectomy**				
no	13 (21.7)	10 (23.8)	23 (22.5)	
yes	47 (78.3)	32 (76.2)	79 (77.5)	0.98
**Completeness of Cancer Resection (CCR)**				
CC0	52 (85.2)	32 (76.2)	84 (81.6)	
CC1	3 (4.9)	4 (9.5)	7 (6.8)	0.59
CC2	4 (6.6)	5 (11.9)	9 (8.7)	
CC3	2 (3.3)	1 (2.4)	3 (2.9)	
CC0/CC1	55 (90.2)	36 (85.7)	91 (88.3)	
CC2/CC3	6 (9.8)	6 (14.3)	12 (11.7)	0.54

PDS: primary debulking surgery NACT-IDS: neoadjuvant chemotherapy and interval debulking surgery CC: completeness of cytoreduction (Harmon R, Sugarbaker P, 2005). CC0: no visible disease, CC1: persisting nodules <0.25cm, CC2: nodules 0.25–2.5cm and CC3: nodules >2.5cm.

**Table 3 cancers-14-02345-t003:** Comparison of the PDS and NACT-IDS populations (complications).

	PDS *n* (%)	NACT-IDS *n* (%)	Total *n* (%)	*p* (Chi2, Yates or Fisher)
**COMPLICATIONS**				
**PER-OPERATIVE**				
no	50 (70.4)	22 (55.0)	72 (64.9)	
yes	21 (29.6)	18 (45.0)	39 (35.1)	0.1
**Severity**				
CTCAE 1–2	18 (100)	12 (92.3)	30 (96.8)	
CTCAE 3–4	0	1 (7.7)	1 (3.2)	0.4
**Transfusion**				
no	39 (84.8)	21 (61.8)	60 (75.0)	
yes	7 (15.2)	13 (38.2)	20 (25.0)	0.018
**EARLY POST-OPERATIVE**				
no	30 (65.2)	24 (61.5)	54 (63.5)	
yes	16 (34.8)	15 (38.5)	31 (36.5)	0.73
**Severity**				
CTCAE 1–2	6 (37.5)	5 (33.3)	11 (34.4)	
CTCAE 3–4	10 (18.8)	10 (66.7)	21 (65.6)	0.64
**LATE POST-OPERATIVE**				
no	39 (73.6)	20 (51.3)	59 (64.1)	
yes	14 (26.4)	19 (48.7)	36 (35.9)	0.03
**Severity**				
CTCAE 1–2	6 (42.8)	7 (41.2)	13 (40.6)	
CTCAE 3–4	8 (57.1)	10 (58.8)	19 (59.3)	1

PDS: primary debulking surgery NACT-IDS: neoadjuvant chemotherapy and interval debulking surgery CC: completeness of cytoreduction (Harmon R, Sugarbaker P, 2005). CC0: no visible disease, CC1: persisting nodules <0.25cm, CC2: nodules 0.25–2.5cm and CC3: nodules >2.5cm. CTCAE: common terminology criteria for adverse events.

**Table 4 cancers-14-02345-t004:** Comparison of the PDS and NACT-IDS populations (adjuvant treatment and follow-up).

	PDS *n* (%)	NACT-IDS *n* (%)	Total *n* (%)	*p* (Chi2, Yates or Fisher)
**Adjuvant chemotherapy**	37 (84.1)	65 (83.3)	102 (83.6)	0.9
**Adjuvant bevacizumab**	11 (18.0)	15 (34.1)	26 (24.8)	0.01
**Adjuvant hormonal therapy**	6 (9.8)	2 (4.5)	8 (7.6)	0.8
**Recurrence or progression**				
no	27 (50.0)	11 (28.2)	38 (40.9)	
Recurrence	20 (37.0)	15 (38.5)	35 (37.6)	0.03
Progression	7 (13.0)	13 (33.3)	20 (21.5)	
total	54	39	93	
**Death**				
no	52 (83.9)	26 (60.5)	78 (74.3)	
yes	10 (16.1)	17 (39.5)	27 (25.7)	0.007
total	62	43	105	

PDS: primary debulking surgery NACT-IDS: neoadjuvant chemotherapy and interval debulking surgery.

**Table 5 cancers-14-02345-t005:** Progression-free survival analysis.

Variables		HR (IC 95%)	*p*
**Age**	<45 y	1	
	>45 y	1.36 (0.82–2.26)	0.23
**NACT-IDS**	no	1	
	yes	1.66 (1.03–2.69)	0.04
**Surgery**	no	1	
	yes	0.64 (0.37–1.11)	0.11
**Peritoneal cytology**	negative	1	
	positive	1.15 (0.59–2.23)	0.68
**Completeness of Cancer Resection (CCR)**	CC0	1	
	CC1	0.81 (0.33–1.97)	
	CC2	2.59 (1.28–5.25)	0.032
	CC3	1.88 (0.58–6.14)	
	CC0-CC1	1	
	CC2-CC3	2.44 (1.3–4.58)	0.004
**Lymphadenectomy**	no paraaortic dissection	1	
	paraortic dissection	2.0 (0.94–4.24)	0.07
	no pelvic dissection	1	
	pelvic dissection	1.46 (0.71–3.00)	0.30
**Intraoperative complications**	no	1	
	yes	1 (0.55–1.82)	0.99
**Early post-operative complications**	no	1	
	yes	1.33 (0.72–2.45)	0.36
**Late post-operative complications**	no	1	
	yes	1.11 (0.63–1.95)	0.72
**Hormonal receptors**	ER −	1	
	ER +	0.39 (0.17–0.89)	0.02
	PR −	1	
	PR +	0.8 (0.45–1.42)	0.45
**Adjuvant treatments**	No chemotherapy	1	
	Chemotherapy	0.78 (0.43–1.45)	0.44
	No bevacizumab	1	
	Bevacizumab	0.97 (0.52–1.84)	0.94
	No hormonal therapy	1	
	Hormonal therapy	1.69 (0.86–3.32)	0.13

PDS: primary debulking surgery NACT-IDS: neoadjuvant chemotherapy and interval debulking surgery CC: completeness of cytoreduction (Harmon R, Sugarbaker P, 2005). CC0: no visible disease, CC1: persisting nodules <0.25 cm, CC2: nodules 0.25–2.5 cm and CC3: nodules >2.5 cm CTCAE: common terminology criteria for adverse events ER: estrogen receptor PR: progesterone receptor. NACT = neoadjuvant chemotherapy; CLA = para-aortic lymph lymphadenectomy; CP = pelvic lymphadenectomy; RE = estrogen receptor; RP = progesterone receptor.

**Table 6 cancers-14-02345-t006:** Overall survival analysis.

Variables		HR (IC 95%)	*p*
**Age**	<45 y	1	
	>45 y	2.32 (1.01–5.32)	0.042
**INITIAL DISEASE**			
**NACT**	no	1	
	yes	2.64 (1.37–5.06)	0.003
**Surgery**	no	1	
	yes	0.4 (0.18–0.9)	0.027
**Completeness of Cancer Resection (CCR)**	CC0	1	
	CC1	1.12 (0.33–3.83)	0.369
	CC2	2.32 (0.89–6.01)	0.369
	CC3	1.23 (0.16–9.31)	0.369
	CC0-CC1	1	
	CC2-CC3	2.01 (0.84–4.82)	0.116
**Lymphadenectomy**	No paraaortic dissection	1	
	Paraortic dissection	0.96 (0.39–2.32)	0.924
	No pelvic dissection	1	
	Pelvic dissection	1.18 (0.46–3.04)	0.726
**Intraoperative complications**	no	1	
	yes	1.06 (0.47–2.41)	0.891
**Early post-operative complications**	no	1	
	yes	0.73 (0.29–1.85)	0.505
**Late post-operative complications**	no	1	
	yes	2.08 (0.91–4.72)	0.081
**Hormonal receptors**	ER −	1	
	ER +	0.23 (0.08–0.64)	0.005
	PR −	1	
	PR +	0.54 (0.22–1.36)	0.192
**Adjuvant treatments**	No chemotherapy	1	
	Chemotherapy	0.77 (0.31–1.89)	0.564
	No bevacizumab	1	
	Bevacizumab	0.32 (0.08–1.35)	0.103
	No hormonal therapy	1	
	Hormonal therapy	2.81 (1.15–6.88)	0.024
**Recurrence**	no	1	
	yes	4.98 (2.6–9.53)	<10–3
**RECURRENT DISEASE**			
**NACT-IDS**	no	1	
	yes	1.67 (0.6–4.59)	0.324
**Surgery**	no	1	
	yes	0.45 (0.07–3.02)	0.406
**Completeness of Cancer Resection (CCR)**	CC0	1	
	CC1	11.3 (0.97–131.85)	0.103
	CC2	11.73 (0.61–227.34)	0.103
	CC3	1.85 (0.16–22.07)	0.103
**Intraoperative complications**	no	1	
	yes	0.66 (0.12–3.53)	0.627
**Early post-operative complications**	no	1	
	yes	0.44 (0.05–4.06)	0.471
**Late post-operative complications**	no	1	
	yes	8.5 (0.77–94.23)	0.081
**Adjuvant treatments**	No chemotherapy	1	
	Chemotherapy	1.53 (0.42–5.6)	0.524
	No bevacizumab	1	
	Bevacizumab	0.72 (0.28–1.84)	0.492
	No hormonal therapy	1	
	Hormonal therapy	0.56 (0.25–1.29)	0.174

NACT = neoadjuvant chemotherapy; CLA = para-aortic lymph nodes; CP = pelvic lymph nodes; RE = estrogen receptor; RP = progesterone Receptor.

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
