# Peer review of "Surgical Implications of Advanced Low-Grade Serous Ovarian Cancer: Analysis of the Database of the Tumeurs Malignes Rares Gynécologiques Network"

_cancers, 2022, doi:10.3390/cancers14092345_

Round 1
Reviewer 1 Report
The authors have presented their data on patients they retrospectively selected from national registry. While the patient database makes this report strong, similar studies have been reported with similar conclusions.
The authors do point out the differences between their studies and those published, but a quick PubMed search showed a paper published recently that presents similar results.
LGSOC is a rare but often fatal disease and lack of information leads to even worse outcomes. So publications like these definitely add to the information pool and improve robustness of available data.
My suggestion is to include this publication (attached) and update the Discussion appropriately.

Author Response
We want to thank the reviewer for these comments. This paper is adressing the same issues as us. It reports very interesting results. Most are similar to ours. But the finding of prognostic impact of SCS>4 is new. We added a paragraph about this paper in the discussion section.
Reviewer 2 Report
It is a rare pleasure to recommend the acceptance of a manuscript “as it is”. The work entitled: “Surgical implications of advanced Low-Grade Serous 2 Ovarian Cancer. A study of the GINECO group & the 3 TMRG network” reports on correctly obtained and clinically usable findings. The manuscript is very well written and makes a relevant contribution to the field. Materials and methods are adequately and comprehensively described. The presentation of the results is systematic and transparent. A limitation of the study is the number of eligible patients - 148 with complete clinical data from 281 LGSOC patients registered with the TMRG network, but this shortcoming is independent of the authors. I appreciate the systematic assessment of adverse events. The study expressly provides relevant conclusions, particularly regarding the non-superiority of NACT in advanced stages of LGSOC. The finding that IDS after NACT was associated with more postoperative complications provides useful arguments for a fair choice of treatment options. The discussion and choice of references could be more detailed, but the discussion (as it is presented) meets the minimum requirements for a good discussion. I recommend an immediate publication of this interesting manuscript.
Author Response
We want to thank the reviewer for these encouraging comments.
Round 2
Reviewer 1 Report
The authors have addressed the concerns adequately.